# Exploring the Effect of Functional Diets Containing Phytobiotic Compounds in Whiteleg Shrimp Health: Resistance to Acute Hepatopancreatic Necrotic Disease Caused by *Vibrio parahaemolyticus*

**DOI:** 10.3390/ani13081354

**Published:** 2023-04-15

**Authors:** Carla Hernández-Cabanyero, Esther Carrascosa, Silvia Jiménez, Belén Fouz

**Affiliations:** 1Instituto Universitario de Biotecnología y Biomedicina (BIOTECMED), Universitat de València, Dr. Moliner, 50, 46100 Burjassot, Valencia, Spain; 2IGUSOL ADVANCE, S.A. Pol. Ind. Lentiscares. C/La Losa, 7, 26370 Navarrete, La Rioja, Spain

**Keywords:** AHPND, *Litopenaeus vannamei*, *Vibrio parahaemolyticus*, functional diets, phytobiotics, essential oils, bacterial challenge, qPCR

## Abstract

**Simple Summary:**

This project explores the effect of feeding whiteleg shrimps with functional diets containing phytobiotics in terms of resistance to acute hepatopancreatic necrotic disease caused by *Vibrio parahaemolyticus*. This disease brings economic losses of billions of dollars in shrimp production worldwide every year. Our study is one step forward in reducing the use of antibiotics in the control of bacterial diseases in aquaculture. The threat of multiresistant bacteria is an issue of major concern in the current One Health context (mainly in the case of potentially zoonotic bacteria such as *V. parahaemolyticus*). Here, we provide evidence that feeding shrimps with diets enriched with mixtures of essential oils can be an advantageous strategy to mitigate infectious diseases and, consequently, reduce the rise of antimicrobial resistance of pathogenic bacteria and build an environmentally friendly and sustainable shrimp aquaculture.

**Abstract:**

Acute hepatopancreatic necrosis (AHPND) is an emerging severe disease caused by strains of *Vibrio parahaemolyticus* (*Vp*_AHPND_) in whiteleg shrimp (*Litopenaeus vannamei*). Mitigating its negative impact, and at the same time minimizing antibiotics treatments, is the major challenge in shrimp aquaculture. A sustainable strategy could be to include immunostimulants in diet. Phytobiotics, harmless plant extracts with immunostimulatory and biocidal activities, are promising candidates. In this study, we evaluated the effectiveness of two diets (E and F) supplemented with phytobiotics (functional diets) in terms of protecting shrimp against AHPND. For this purpose, groups of animals were fed functional or control diets for 4 and 5 weeks and, subsequently, they were challenged with *Vp*_AHPND_ by immersion. We compared the mortality in infected groups and estimated the percentage of carriers by using a specific qPCR in hepatopancreas tissue. The results showed that mortality was significantly lower in the group fed functional diet E and, after a 5-week feeding schedule. This group also showed the lowest percentage of carriers. The pathological effects were also reduced with diet F. Thus, feeding shrimp with phytobiotic-enriched diets in critical periods will be highly beneficial because it increases the host’s resistance to AHPND pathology.

## 1. Introduction

The whiteleg shrimp (*Litopenaeus vannamei*) is a native species of the eastern coast of the Pacific Ocean. Tropical marine environments (from the Mexican state of Sonora to the Peruvian city of Tumbres), where the water temperature is above 20 °C all year round, are its natural habitats [1]. In 1973, artificial propagation of this species was achieved for the first time in Florida. Consequently, the production transitioned from traditional wild fishing to controlled culture [2]. *L. vannamei* production has grown exponentially in the last two decades and currently accounts for 52.9% of crustacean production worldwide [3]. In fact, it is the most widely cultured shrimp species due to its good taste, rapid growth, and excellent saline adaptability [1]. Infectious diseases are the major threat to *L. vannamei* production, with acute hepatopancreatic necrotic disease (AHPND) being one of the most limiting. In fact, it has caused economic losses of billions of dollars since 2010, when it was first detected in China and Vietnam [3,4]. Since its emergence, it has rapidly spread and cases have been reported in Malaysia (2010), Thailand (2011), Mexico (2013), Philippines (2014), South American Countries (2014–2016), Bangladesh (2017), the USA (2017), Taiwan Province of China (2018), South Korea (2019), and the Okinawa Prefecture of Japan (2020) [5,6,7,8,9,10,11,12,13,14,15].

The etiological agent of the disease is *Vibrio parahaemolyticus*, a Gram-negative bacterial species inhabiting aquatic saline ecosystems [16]. It encompasses strains with a high metabolic diversity, able to survive under conditions of different temperature, salinity, and pH in marine and estuarine environments where shrimp farms are housed [13]. Virulent *V. parahaemolyticus* strains specifically contain the pVA1 plasmid (*Vp*_AHPND_ strains), which encodes the PirA and PirB toxins, whose cytotoxic effect consists of the formation of pores in the membranes of its target cells, leading to their death by osmotic lysis [4,17]. In fact, knock-out mutants of the *pirA* and *pirB* genes do not produce toxins and are not able to induce the AHPND in healthy animals [18]. Infection of shrimps with *Vp*_AHPND_ occurs mostly via the oral route [4]. In breeding tanks, these bacteria can be found suspended in water or forming biofilms in sediments. Other sources of the pathogen would be feed or contaminated fresh food, as well as sick animals (shrimps are cannibalistic animals) [4]. When a healthy shrimp ingests *Vp*_AHPND_, the bacteria enter the digestive tract, colonizes the stomach, and releases PirA and PirB toxins to the hepatopancreas (HP) [4]. The HP tubules contain a layer of epithelial tissue that carries out digestive enzyme secretion and nutrient absorption and storage functions [19]. The toxins induce necrosis of these epithelial cells, leading to degeneration of the tubules which, in an advanced stage of the disease, results in dysfunction of the HP and ultimately death of the animal [4]. The outcome of AHPND results is a mortality rate of up to 100% within 30–35 days of introducing post-larval or juvenile shrimp into contaminated rearing tanks [20]. Diagnosis is based on the observation of clinical signs (loss of appetite, empty stomach, and whitish and atrophic HP), histological preparations (necrosis of HP epithelial cells, hemocytic infiltration, and presence of bacteria), and molecular techniques (detection of *pirA*_vp_ and *pirB*_vp_ genes by conventional or quantitative PCR (qPCR)) [4].

Although originally, only *Vp*_AHPND_ strains were responsible for the pathology, AHPND should be considered an emerging disease since accumulating evidence shows that it can also be attributed to other *Vibrio* species such as *V. harveyi* [21], *V. owensii* [22,23], *V. campbellii* [24,25], and *V. punensis* [26], all of which harbor the *pirA* and *pirB* genes on plasmids homologous to pVA1. These genes have been acquired by conjugative horizontal transfer and contribute to increasing the diversity of the strains causing AHPND and the risk for shrimp production [27,28].

For all the above-mentioned reasons, preventing this disease or mitigating its negative impact in shrimp aquaculture is a priority. However, on the one hand, vaccines in invertebrates are not an option, due to the primitive immune system of the animals [29]; on the other hand, regulations continue to be more restrictive to avoid public health concerns caused by the use of antibiotics (emergence of resistant bacteria), the most extended therapeutic practice against the AHPND [29]. In the current One Health context, it is urgent to find alternative measures which are safe and environmentally and economically sustainable. Different strategies have been proposed, such as the promotion of shrimp pregrown in biofloc systems [28], the use of oxygen or ozone nanobubbles [30], phage-therapy [31], the addition of microalgal–bacterial consortia [32], and probiotics [33,34,35]. However, there is little information regarding the effects of functional feed additives for controlling shrimp bacterial diseases [36,37,38]. In recent studies, shrimp fed diets containing ethanol extracts of turmeric (*Curcuma longa*), maca (*Lepidium meyenii*), and ginger *(Zingiber officenale*) inhibited the growth of *Vibrio* species and the formation of biofilms of *V. parahaemolyticus* [39,40]. In addition, other authors demonstrated that specific molecules such as vitamin C, essential aminoacids (arginine), and sodium ascorbate improve the immune response in shrimp and confer protection against *Vibrio* infection [18,36,39,41].

Some of these studies support the idea that phytobiotic compounds could be promising candidates since they are plant extracts containing harmless compounds with immunostimulatory and biocidal activities [42]. Moreover, they are not expensive, as the production process to obtain plant extracts is generally simple, and easy to administer orally by introducing them into the food. In terms of safety, they are environmentally friendly because they are highly biodegradable and the likelihood of generating resistant microorganisms is very low as they do not contain single active molecules directed at a specific cellular target but usually include a wide range of bioactive ones [43,44,45].

In this work, two diets supplemented with phytobiotic compounds (mixtures of essential oils) have been administered to shrimp to evaluate their putative beneficial effect on the resistance of the animals to the AHPND pathology. We compared the mortality of the groups fed the functional diets for 4 and 5 weeks with that of the control group after an induced infection with *Vp*_AHPND_ strain. Additionally, we calculated the percentage of carriers and the bacterial load in HP of survivors by using an optimized qPCR protocol. Since there are hardly any reports on whether the enrichment of diets with phytobiotics can effectively improve the health of *L. vannamei,* this study will represent an advance in the fight against the AHPND.

## 2. Materials and Methods

A schematic overview of the experimental design is represented in Figure 1.

### 2.1. Bacterial Strain and Growth Conditions

*Vp*_AHPND_ strain, provided by Lee et al. [18], was used for in vitro and in vivo assays. Other *Vibrio* strains were also included in PCR assays (Appendix A). Bacteria were routinely grown on Trypticase Soy agar or broth, both supplemented with 1% NaCl (TSA-1 or TSB-1, respectively) at 25 °C for 24 h and stored in Luria Broth-1 plus 20% glycerol at −80 °C.

### 2.2. Additives: In Vitro Study

The phytobiotic additives, provided by the Spanish company IGUSOL ADVANCE S.A., consisted of microencapsulated mixtures of essential oils from thyme and cinnamon (additive E) and from oregano and cloves (additive F), with each capsule (size from 0.1–0.2 mm to 0.3–0.4 mm) containing 30% of essential oils. The precise composition of the essential oil blends remains confidential as these additives are in the final stages of development and license. The antibacterial activity against *Vp*_AHPND_ of the additives was evaluated by determining the minimum inhibitory concentration (MIC) and the minimum bactericidal concentration (MBC) according to Alderman and Smith [46]. Briefly, stock solutions of the additives were prepared in Brain–Heart Infusion (BHI) supplemented with 1% NaCl (BHI-1) at a final concentration of 2500 ppm and filtered (0.22 mm), and different dilutions were prepared (100–1000 ppm). Bacterial culture in BHI-1 at stationary phase was used to inoculate (10^6^ CFU/mL) tubes containing the diluted additives (5 mL). Bacterial growth controls were performed in a medium without additive (positive) and a medium with 4 µg/mL tetracycline, concentration (negative) (Clinical and Laboratory Standards Institute, CLSI, guidelines). After incubation at 25 °C for up to 5 days, MIC was established as the lowest additive concentration inhibiting bacterial growth (without turbidity). A total of 10 μL aliquots from tubes without turbidity were drop-plated on TSA-1 and incubated at 25 °C for 48 h. MBC was determined as the lowest additive concentration at which no bacterial colonies grew. Ten independent technical replicates were performed for each condition.

### 2.3. Functional Diets

Diets used in this study were prepared by cooking extrusion with a semi-industrial twin-screw extruder (CLEXTRAL BC-45, Firminy) in the pilot plant of the Polytechnic University of Valencia. The processing conditions were as follows: 55–60 rpm speed screw, temperature of 100 °C, 40–50 atm pressure, and pellets with diameters of 3 mm. Diets yielded around 35% crude protein and 20 MJ/Kg of energy contribution. Control diet (CTRL, diet G) contained the basal ingredients (Table 1), and functional diets (E and F) were supplemented with phytobiotic additives (diet E with 0.9 g of additive E/Kg and diet F with 0.8 g of additive F/Kg). Shrimps were daily fed CTRL diet during the acclimation period and CTRL or functional diet during the feeding trials (4–5 weeks).

### 2.4. Animal Maintenance and Feeding Schedules

Shrimps used in this study were purchased from the company, White Panther (Republic of Austria). Animals (average weight of 0.4 g) were acclimatized for one month and maintained in the Fish facilities (code ES461900001203) of the Central Service for Experimental Research (SCSIE) of the University of Valencia (Spain) in 200 L tanks containing 150 L of saline water (34 ppt salinity, 25 °C, pH 7.8–8.2) with a ratio of 35 animals/tank. Water quality parameters (pH, temperature, salinity, and nitrogen compounds) were controlled daily. Shrimps were fed CTRL diet over the course of this period until they reached an average weight of 1 g. Then, the feeding trials with functional diets started. For this purpose, animals were distributed in 3 groups of 42–48 animals, 1 group fed CTRL diet and the other groups fed functional diets (E and F) for 4–5 weeks with a daily regimen of around 10% of the shrimps’ weight. Shrimp were fed by hand three times a day (9:00, 13:00 and 17:00), distributing pellets slowly in order for all animals to eat.

### 2.5. Optimization of the Infection Model and Challenge Dose Validation

Prior to in vivo experiments, it was verified that *V. parahaemolyticus* was not part of the endogenous microbiota of the shrimps by sampling HP of 6 randomly selected animals from the control population on Thiosulphate Citrate Bile Saccharose (TCBS) agar (Condalab), a selective and differential medium for vibrionaceous bacteria, and then identifying the colonies grown by phenotypic tests. *V. parahaemolyticus* grows by forming big green colonies on TCBS agar.

Bath-immersion was selected as the optimal infection model since it simulates the natural transmission conditions of *V. parahaemolyticus* [18,47,48]. Exposure time (bath length) and different bacterial concentrations were tested in a pre-challenge experiment to validate bath conditions and a suitable infective dose for the bacterial challenge. Infections were carried out in 200 L tanks and the water was maintained at the same conditions as in the maintenance tanks (see Section 2.4). Subgroups of 4–5 animals (1–2 g average) fed CTRL diet were exposed to a range of *Vp*_AHPND_ strain concentrations (10^5^–10^8^ CFU/mL of water). Negative controls (mock groups) were immersed in TSB-1 culture medium at the same dilution. Then, animals were transferred into new tanks containing clean water and kept under constant conditions. The animals were monitored for the appearance of clinical signs and mortality was recorded for 6 days post-challenge (p.ch.). Moribund and/or dead animals were removed daily, and HP was sampled by streaking on TCBS agar. Mortalities were considered to be due to *V. parahaemolyticus* only if the bacterium was recovered in pure culture from HP. For identification of the pathogen, a slide agglutination test with the corresponding anti-plasma was used. The dose producing mortalities between 40 and 50% (LD_40–50_) was chosen for the induced infections. All experiments were performed in triplicate.

### 2.6. Bacterial Challenge

At the end of the feeding periods (4 and 5 weeks), groups of 12–16 shrimps (2 g average) per dietary treatment were bath challenged with the previously determined dose of *Vp*_AHPND_ and optimized conditions and transferred into new 200 L tanks. A group of 12 animals per treatment was challenged with TSB-1 as a control of the experimental handling. Each shrimp group were fed the same diet (either CTRL or functional) during the post-challenge period and during the feeding trial. Animals were monitored for the appearance of clinical signs and mortality was recorded daily for 6 days p.ch.. Moribund and/or dead animals were removed daily from the tanks and HP was sampled and streaked on TCBS to confirm the causative agent of mortality. In all challenges, shrimps showing signs of disease (animal near the water surface, slow swimming, or motionless at the bottom of the tank) were sacrificed by overexposure to the anesthetic as mentioned below. Post-mortem examination was performed as described above. Three independent replicates (infection tanks) were performed for each condition (diet and feeding time).

### 2.7. Sample Collection

In all lethal samplings, shrimps were decapitated after 3-aminobenzoic acid ethyl ester (MS-222, 100 μg/mL) over-exposure, and all efforts were made to minimize suffering.

Moribund or recently dead shrimps were removed from the tanks and observed for external and internal clinical signs. After necropsy, HP was collected and, after streaking on TCBS agar, was stored at −80 °C for future DNA extraction and qPCR analysis.

Survivor shrimps were sacrificed at 6 days p.ch. and HP was taken and processed as in dead shrimps.

Tank water was sampled to detect the pathogen. A total of 50 mL of water were daily collected from representative tanks from each group along 3 days p.ch. Samples were concentrated by centrifugation at 5,300 rpm for 30 min and resuspended in 1 mL of sterile phosphate buffered saline (PBS, pH 7.6). A second centrifugation step at 17,000 rpm for 10 min was performed before resuspending the pellet in 100 µL PBS. Samples were used for DNA extraction and qPCR as specified in the following sections.

### 2.8. DNA Extraction from Shrimp and Water Samples

DNA extraction from shrimp HP was performed using the Genomic DNA Purification Kit (ThermoFisher, Waltham, MA, USA) according to the manufacturer’s specifications. HP was weighted before the DNA extraction, and the bacterial load was normalized according to the HP size. Then, under sterile conditions, the organ (0.02–0.04 g) was homogenized in 200 µL Tris-EDTA buffer and 400 µL of the lysis solution was added. This mixture was then incubated in a bath at 65 °C for 15 min, inverting the tubes every 3 min. Then, DNA was purified by adding 600 µL of chloroform, emulsifying by inversion and centrifugating at 10,000 rpm for 2 min. A total of 400 uL of the supernatant (DNA) was collected and transferred to an eppendorf tube containing 800 µL of precipitation solution. After 1 min at room temperature, the tubes were centrifuged at 10,000 rpm for 2 min. The supernatant was removed, and the pellet was resuspended in 100 µL of buffered saline solution. Finally, DNA was precipitated, adding 300 µL ice cold absolute ethanol and incubating for 30 min at −20 °C. After this time, the samples were centrifuged at 10,000 rpm for 10 min, the supernatant was discarded, and the DNA resuspended with 300 µL of 70% ethanol and centrifuged again at 10,000 rpm for 10 min. The supernatant was discarded, and the purified DNA pellet was incubated at RT until it was completely dried and resuspended in 50 µL of Nuclease Free Water (Invitrogene, Waltham, MA, USA). DNA quality and yield of the extraction was checked using a NanoDrop2000 spectrophotometer (ThermoFisher). DNA was stored at −20 °C until use.

DNA from water samples was extracted by boiling method. Briefly, the sample was incubated in a water bath at 100 °C for 10 min, followed by ice incubation for 5 min and centrifuged for 5 min at 17,000 rpm. The supernatant was transferred to an eppendorf tube and stored at −20 °C until use.

### 2.9. qPCR Specific for V. parahaemolyticus (Vp qPCR) Detection and Quantification

For the detection and quantification of *Vp*_AHPND_ (in shrimp and water samples), a specific qPCR (*Vp* qPCR) was optimized. The specificity of several primers (Appendix A) was tested using representative *Vibrio* species (*V. alginolyticus, V. harveyi* and *V. vulnificus*) and several *V. parahaemolyticus* strains (*Vp*_AHPND_, Vp lab1, Vp lab2 and CECT 8407) from our laboratory or the Spanish Type Culture Collections (Appendix A). The detection limit was established using decreasing concentrations of *Vp*_AHPND_ strain DNA (150,000 pg/µL–0 pg/µL DNA) and decreasing concentrations of bacteria (1 × 10^8^ CFU/mL–0 CFU/mL). Different programs were tested on the StepOnePlus™ Real-Time PCR thermal cycler (Agilent Technologies) and reagents and volumes were optimized for the qPCR reaction. After optimization, the qPCR reaction was performed using Power SYBR^®^ green PCR Mastermix reagent (Invitrogen) in a final volume of 10 µL/sample (2 µL nuclease free water, 1 µL 100 µM forward primer and 1 µL 100 µM reverse primer, 5 µL SYBR Green, 1 µL DNA). The amplification conditions selected were as follows: stage 1, (initial or sustained stage): 95 °C for 5 min; stage 2, (cyclic step × 40 cycles): 95 °C for 15 s, 60 °C for 1 min; and stage 3, (melting curve stage): 95 °C for 15 s, 60 °C for 1 min, 95 °C for 15 s. Cycle threshold (CT) values were determined with StepOne v. 2.0 software.

### 2.10. Statistical Analysis

Differences in the probability of death between the groups fed functional diets and control diet (diet effect), as well as between the two challenges (time effect), were identified by performing logistic regression analysis using the RStudio interface.

## 3. Results

### 3.1. In Vitro Characterization of the Antibacterial Activity of the Additives

The antibacterial activity of the phytobiotic additives was evaluated by determining the MIC and MBC against *Vp*_AHPND_ in vitro. An initial screening of concentrations between 100 and 1000 ppm showed that both MIC and MBC were 200 ppm for additive E and 300 ppm for additive F. In a second more precise screening, additives E and F showed MIC and MBC values of 195 ppm and 220 ppm, respectively. Both phytobiotic additives showed bactericidal activity against *Vp*_AHPND_ at low concentration (around 200 ppm); thus, they could be considered as suitable candidates to supplement diets.

### 3.2. Infection Model and Challenge Dose

The infection model was optimized by immersing groups of 4–5 animals at different bacterial concentrations for 30 or 60 min. A first challenge, using 1 g average shrimps, evidenced that mortality only occurred when the exposure time was 1 h and the bacterial dose higher than 4 × 10^6^ CFU/mL, achieving 100% mortality when the dose was 5 × 10^7^ CFU/mL (Table 2). A second challenge, using 2 g average shrimps, showed that these were more resistant, with LD_50_ dose of around 5 × 10^7^ CFU/mL (Table 2). In all cases, mortality was observed between 24–72 h p.ch. The presence of *Vp*_AHPND_ in moribund/dead animals was confirmed by isolation from HP on TCBS agar. As expected, no mortality was observed in non-infected mock groups. Immersion in baths containing *Vp*_AHPND_ at 5 × 10^7^ CFU/mL for 1 h were the selected conditions for bacterial challenges at the end of the feeding trials.

### 3.3. Optimization of the Vp qPCR

In order to optimize the qPCR protocol to detect and quantify *Vp*_AHPND_ in animals after being challenged, we first tested the specificity of several primer pairs against representative strains of *V. parahaemolitycus*. All the primers except VPF/VPR (which target the *tlh* gene, encoding the species-specific thermolabile hemolysin [49]) and PirA-F/PirA-R (which target the *pirA* gene, encoding the PirA toxin [50]) gave multiple specificities as well as positive identification of other *Vibrio* species (mainly *V. vulnificus*). Moreover, VPF/VPR primers were more sensitive than the PyrA-F/PyrA-R (Appendix A). Although VPF/VPR primers do not allow us to differentiate virulent and non-virulent strains (Appendix A), challenges were carried out under controlled conditions, so we prioritized sensitivity over strain specificity.

The average CT values using VPF/VPR primers to amplify *V. parahaemolyticus* DNA (1 µg/µL) were very low (12.32) (Table 3), while those obtained using DNA from selected *Vibrio* species were similar to those of the negative control (around 33) (Table 3).

Sensitivity of selected primers was assessed using purified *V. parahaemolyticus* DNA and bacterial culture, with the lowest concentrations being detected at 0.15 pg/µL and 1 CFU/mL, respectively (Figure 2). These values were established as the detection limits of the *Vp* qPCR protocol.

### 3.4. Effect of the Functional Diets in Protection against Vp_AHPND_

First, we confirmed the absence of *V. parahaemolyticus* in commensal microbiota from HP of shrimps. After purification, the isolated colony types on TCBS agar were identified as *V. fluvialis* or *V.fluvialis*-like species using the phenotypic API20E system (profiles 2042124 and 3042025, with a probability of identification 98.6–98.9%, and 3044125, with a probability of identification 57.5%). Then, bacterial challenges were carried out at the end of the feeding trials. The infective doses were 6 × 10^7^ CFU/mL (4-week feeding challenge) and 4.2 × 10^7^ CFU/mL (5-week feeding challenge), both close to the previously determined LD_50_ (5 × 10^7^ CFU/mL).

#### 3.4.1. Mortality in Bacterial Challenges

Clinical signs and mortality (Table 4) were monitored and recorded daily for 6 days p.ch.. *Vp*_AHPND_ was isolated from HP of all moribund or recently dead animals on TCBS agar.

In the two bacterial challenges, mortality in group/tank E3 reached 100% in the first 24 h. p.ch. (Table 4). In both cases, *Vp*_AHPND_ was not isolated from HP of the dead animals and the CT values in *Vp* qPCR were very high (between 31 and 33), equivalent to bacterial loads <10 CFU/g (considered as non-carriers, see Table 5). Given these results, we suspected that the cause of mortality was not the bacterial pathogen but that some environmental disturbance. Water quality parameters (pH, temperature, salinity, and nitrogen compounds) presented no deviation in any of the experimental tanks, so we hypothesized that some toxic debris (i.e., detergent debris) present in that tank could have altered the physiological state of the shrimps and caused their death. A subsequent bath with a group of 6 shrimps in the same tank E3, without *Vp*_AHPND,_ resulted in the death of all animals, supporting our hypothesis. For these reasons, we did not consider the results of the group in tank E3 in the final analysis.

In the first trial, the mortality of the CTRL group was close to 50%, but in the second trial, it rose to almost 70% (Figure 3 and Figure 4). The cumulative mortalities and percentage of survival in shrimps along the challenge is represented in Figure 3 and the average percentage of final mortality for each group in the two challenges is graphically represented in Figure 4. Mortality in both challenges was clearly lower in the group fed the functional diet E, with 15.63% (4-week feeding) and 29.67% (5-week feeding) on average (Figure 3 and Figure 4). In the groups fed the CTRL diet and the functional diet F, mortality was higher (between 40–50% and 55–70% after 4- and 5-week feeding, respectively). In the case of shrimp fed diet F, only after 5-week of administration did the survival of animals improve compared to that of the CTRL group (Figure 3 and Figure 4).

Challenge data (excluding the group maintained in E3 tank) were subjected to logistic regression analysis. The “casualty” and “survivor” data of the shrimp fed CTRL diet G in challenge 1 (G1 tank) were used as reference for comparison, with the mortality of this group represented in the intercept of the model. The results of the analysis showed significant differences in the probability of death in the group fed functional diet E challenged after 4- or 5-week feeding with respect to the intercept.

The presence of *Vp*_AHPND_ in HP from moribund animals was confirmed and quantified by *Vp* qPCR, obtaining in all cases CT values between 14 and 27, equivalent to bacterial concentrations between 5 × 10^3^ and 1 × 10^9^ CFU/g of HP (Table 5).

#### 3.4.2. Bacterial Loads in Survivors

6 days p.ch. HP of survivor shrimps were sampled on TCBS agar, and DNA was extracted and subjected to *Vp* qPCR. All animals with positive *Vp*_AHPND_ colonies on TCBS showed positive values in qPCR (CT values ranging between 14–33) (carrier animals) (Table 5). The HP samples with CT values higher than 33 did not yield colonies on TCBS agar; therefore, they were considered non-carriers (Table 5). The CT values obtained were interpolated to the standard curve to obtain the corresponding CFU/g. Based on the bacterial load detected in HP of survivors, different carrier categories (high, medium, or low) were established (Table 5).

The bacterial load in HP of survivors varied between animals fed different diets and challenged after 4 or 5 weeks of feeding (Table 6). Animals fed the functional diet F showed slightly higher CT values and lower bacterial loads than those fed the functional diet E (Table 6).

The average percentage of carrier animals after bacterial challenge showed high variability among subgroups fed the same diet (Figure 5). In the first challenge, similar average percentages were obtained in all groups, with a slight decrease in those fed functional diets, although it was not statistically significant (Figure 5). In the second challenge, a high statistically significant difference was observed in the group fed diet E, which showed only 20% of carriers compared to around 60% in the CTRL group (Figure 5).

### 3.5. Bacterial Loads in Water

DNA obtained from water samples of the tanks in which animals were maintained after the induced infections was subjected to *Vp* qPCR. Only those water samples analyzed before the challenges were negative for *V. parahaemolyticus* (Table 7). All samples taken after challenge showed CT values between 22 and 29, which correspond to bacterial concentrations between 5 × 10^3^ and 5 × 10^8^ CFU/mL (Table 7). These values were relatively stable during 3 days p.ch. (Table 7).

## 4. Discussion

AHPND is a devastating emerging bacterial disease that affects the shrimp aquaculture industry, causing losses amounting to billions of dollars [51]. According to the Aquatic Animal Health Code of the World Organization for Animal Health (WOAH), the whiteleg shrimp is the most susceptible species to this disease [52]. Even if *V. parahaeomlyticus* is considered the main etiologic agent, other *Vibrio* species have been associated with the disease [53,54]. These findings highlight the importance of shrimp farms as hot spots for *Vibrio* spp. evolution through the acquisition of virulent traits that will favor the spread of the disease. Although the zoonotic potential of *V. parahaemolyticus* remains to be clarified, zoonosis due to other vibrios have been reported [55]. Therefore, the spread of the disease should be considered not only a problem for aquaculture sustainability but also a public health concern. Preventive therapies such as vaccines have very limited success to control AHPND [56,57]; thus, antibiotics have been traditionally used for this purpose. In 2017, 93.8% of antibiotics consumption worldwide occurred in Asia and 2.7% was associated with the shrimp industry [58,59]. Therefore, it is necessary to find cost-effective, easy-to-manage, and safe alternatives to antibiotics in response to bacterial diseases. Phytobiotics are gaining attention as a promising option for controlling infections in shrimp [60,61,62,63].

To gain knowledge in this field, we first evaluated the benefits of selected phytobiotic additives (mixtures of essential oils from thyme and cinnamon [E] or oregano and cloves [F]) and confirmed they have a bactericidal effect in low concentrations, as observed in other plant-derived compounds (turmeric and ginger) and seaweeds [37,38,40,62]. From these promising in vitro results, we developed diets enriched with both additives and assessed their effect on shrimp resistance to AHPND. Induced infections with *Vp*_AHPND_ strain in shrimps fed the functional diets for 4 or 5 weeks showed conclusive and significant results, despite the high standard deviation (probably) due to the variability implicit in the population used (they are not pure genetic lines). Diet supplemented with the additive E and administered for 4 weeks was highly beneficial for shrimp health, reducing mortality up to 15.63% compared to the 40% observed in the CTRL group. Diet F, however, needs to be administered for at least 5 weeks to improve shrimp survival after bacterial challenge. Our results are in line with those recently reported by Quiroz-Guzmán et al. [39], who observed that after 6 weeks of feeding with functional diets containing curcuma and maca or vitamin C, shrimps showed significantly higher survival rates (85%) after bacterial challenge compared to that of the control groups (50%–55%). The strength of our results rests on the shorter feeding schedule with functional diet E (4 weeks vs. 6 weeks in the Quiroz-Guzmán study [39]), providing similar protective effect against AHPND. The drop of the shrimp survival after the 5-week feeding schedule compared to the 4-week schedule could indicate that animals were more vulnerable at that time for unknown reasons that might be investigated in future studies. Differences in mortality observed between our study and that of Quiroz-Guzmán et al. [39] could also be due to the challenge model; they used an injection method, and we selected bath immersion, simulating the natural mode of transmission of the disease. This fact highlights the importance of standardizing infection methods in order to compare study results easily and reliably.

Interestingly, the same authors suggested that the beneficial effect of TuMA diet (containing turmeric and maca) could be due to the combined antibacterial properties against Vibrionales, especially *V. parahaemolyticus*, and the promotion of a desirable bacterial community in the shrimp intestine [39]. We did not study how functional diets affected the gut microbiota composition; but we assessed another approach, the *V. parahaemolyticus* loads in HP of survivors after induced infection. The percentage of carriers was lower in animals fed functional diets for 4 weeks than in those fed the CTRL diet, and the significant lowest ones were found in group fed diet E for 5 weeks, demonstrating its extra benefit reducing the spread of the disease. Therefore, diet E showed the most promising results in all experimental approaches addressed, reinforcing itself as a good candidate to mitigate the effects of AHPND in shrimp culture. Whether the lower percentage of carrier animals was due to an impaired bacterial colonization or to a faster pathogenic bacterial elimination requires further investigation.

Finally, we demonstrated the presence of *V. parahaemolyticus* in the water of the tanks maintaining infected shrimp throughout the post-challenge period, which would indicate that the animals are continuously releasing the bacterium into the environment. In fact, the *V. parahaemolyticus* population in water remained constant for 3 days p.ch. Although longer follow up would be necessary to determine if bacterial loads decrease over time, our observations suggest that carrier asymptomatic animals represent a risk of disease outbreaks since they favor the accumulation of the pathogen on water. In any case, the pathogen could be quantified in water without killing animals by using our optimized *Vp* qPCR, a valuable tool to detect risk of outbreaks and take adequate measures to prevent them.

It is important to remark that we have adapted the time of functional diet administration to the highly risky period for disease outbreaks in farms, often within 30–35 days after introducing post-larval or juvenile shrimp into contaminated tanks. Nevertheless, further studies are needed to assess the effectiveness of the additive dose and the feeding regime used in this study when scaling to shrimp farm conditions.

## 5. Conclusions

In this study, the diet supplemented with a mixture of essential oils from thyme and cinnamon (phytobiotic additive E) significantly improved the resistance of *L. vannamei* to AHPND pathology. Regression analysis indicated that the survival of shrimps challenged with *V. parahaemolyticus* increased if they were fed a phytobiotic-enriched diet for 4 or 5 weeks. Moreover, this feeding strategy also promoted the reduction of the percentage of carriers after the induced infection.

The administration of a diet supplemented with phytobiotic E for 4 weeks in critical periods, i.e., just after introducing shrimps into rearing tanks or before stressful events, could significantly benefit animals by mitigating the negative effects of the AHPND and, in all likelihood, other pathologies.

## Figures and Tables

**Figure 1 animals-13-01354-f001:**
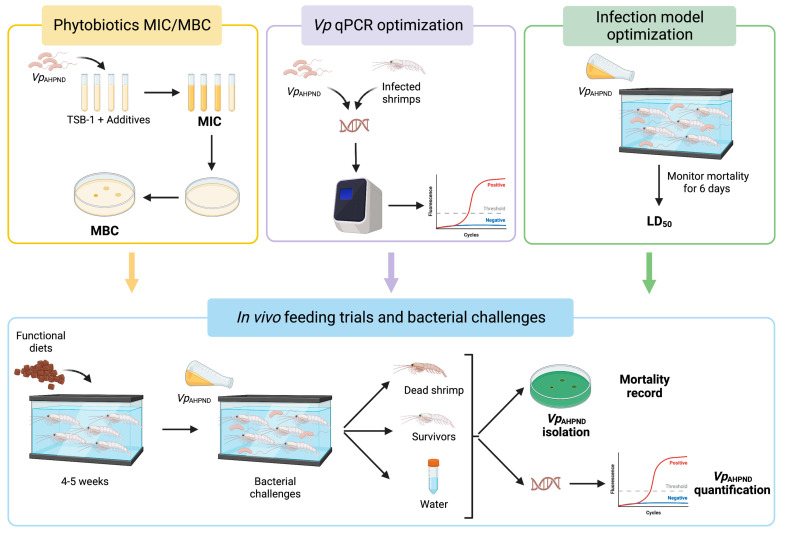
Overview of the experimental design of this study. Created with BioRender.com.

**Figure 2 animals-13-01354-f002:**
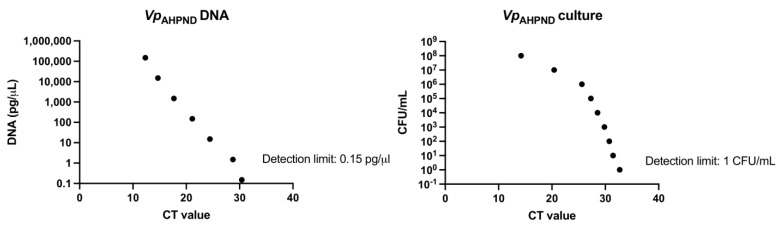
*Vp* qPCR sensitivity and detection limit. *Vp* qPCR sensitivity was tested using both purified DNA and bacterial cells and the detection limits were established. The figure shows the standard curves built from the CT values obtained in *Vp* qPCR of samples with different concentrations of DNA (left) and bacterial cells (right) of the *Vp*_AHPND_ strain.

**Figure 3 animals-13-01354-f003:**
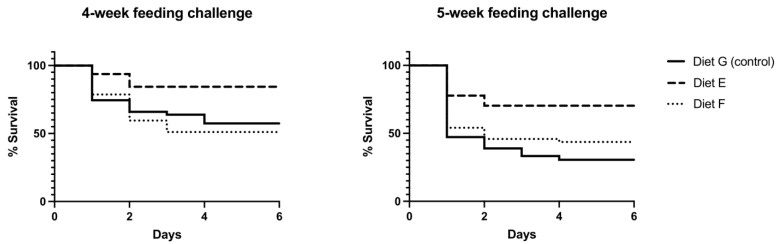
Cumulative mortalities and percentage of survival in shrimps fed functional diets E and F or control diet G (for 4 or 5 weeks) and challenged with *Vibrio parahaemolyticus* (*Vp*_AHPND_ strain).

**Figure 4 animals-13-01354-f004:**
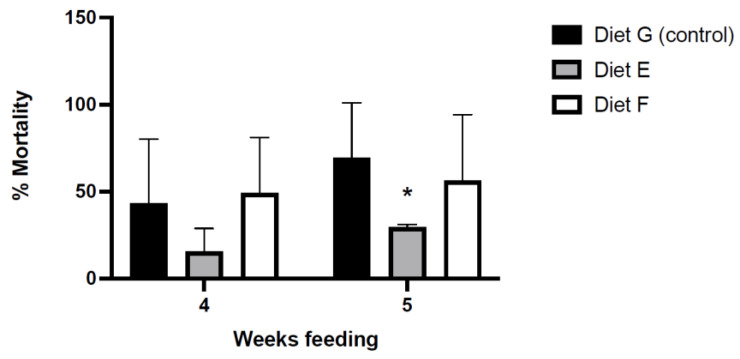
Average percentage of mortality in shrimps fed functional diets (for 4 or 5 weeks) and challenged with *Vibrio parahaemolyticus* (*Vp*_AHPND_ strain). *: significant differences in mortality (logistic regression analysis). The “casualty” and “survivor” data of the shrimp fed control diet G in challenge 1 (G1 tank) were used as reference for comparison, with the mortality of this group represented in the intercept of the model.

**Figure 5 animals-13-01354-f005:**
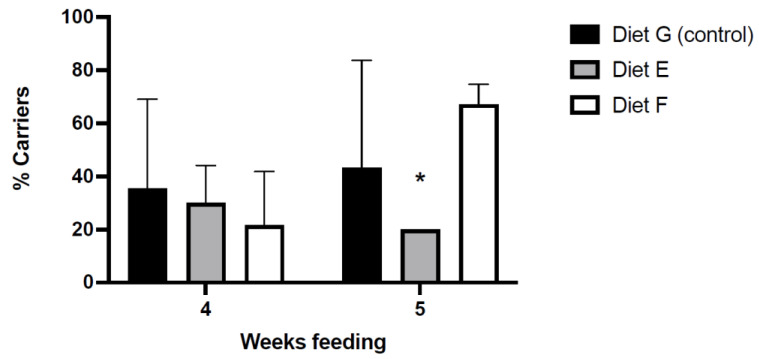
Average percentage of carrier animals among survivors fed functional diets (for 4 or 5 weeks) and challenged with *Vibrio parahaemolyticus* (*Vp*_AHPND_ strain). *: significant differences in mortality in group fed functional diet vs. group fed control diet (diet G).

**Table 1 animals-13-01354-t001:** Ingredient content and proximate basal composition of experimental diets.

Ingredient	g/Kg
Fish flour	150
Soybean	250
Gluten	90
Wheat	375
Soybean oil	32.5
Fish oil	32.5
Calcium phosphate	25
Lecithin	15
Maltodextrin	20
Vitamin supplements	10

**Table 2 animals-13-01354-t002:** Conditions assessed in the bath challenge model of shrimp with *Vibrio parahaemolyticus* (*Vp*_AHPND_ strain).

Shrimp Average Weight (g)	Bacterial Dose (CFU/mL)	Bath Length (min)	Mortality (%)
1	5 × 10^7^	30	0
1	5 × 10^7^	60	100
1	2 × 10^7^	60	75
1	4 × 10^6^	60	0
1	None	60	0
2	5 × 10^7^	60	50
2	1 × 10^7^	60	0
2	8 × 10^6^	60	0
2	None	60	0

**Table 3 animals-13-01354-t003:** Results of qPCR with VPF/VPR primers for *Vibrio parahaemolyticus* detection (*Vp* qPCR) using DNA samples from a range of *Vibrio* species.

DNA from	CT Value (*Vp* qPCR)
*V. parahaemolyticus*	12.32
*V. harveyi*	35.56
*V. vulnificus*	34.48
*V. alginolyticus*	28.25
Negative control (H_2_O)	33.12

**Table 4 animals-13-01354-t004:** Mortality of shrimps fed functional diets for 4–5 weeks and challenged by immersion with *Vibrio parahaemolyticus* (*Vp*_AHPND_ strain).

Challenge with *Vp*_AHPND_ after	Diet	Tank	Total Animals (n)	Dead Animals (n)
4-week feeding schedule	G (control)	G1	16	7
		G2	16	1
		G3	15	12
	E (functional)	E1	16	4
		E2	16	1
		E3	16	16
	F (functional)	F1	16	2
		F2	15	10
		F3	16	11
5-week feeding schedule	G (control)	G1	12	4
		G2	12	10
		G3	12	11
	E (functional)	E1	13	4
		E2	14	4
		E3	14	14
	F (functional)	F1	16	12
		F2	16	13
		F3	16	2

**Table 5 animals-13-01354-t005:** Categories of carrier animals according to *Vibrio parahaemolyticus* load in hepatopancreas (HP). CT range and equivalences in CFU/g in HP from diseased/dead and survivor animals.

Type of Animal	CT Range (*Vp* qPCR) *	*Vibrio parahaemolyticus* (CFU/g HP) *	Carrier Category
Dead (diseased)	14–27	5 × 10^3^–1 × 10^9^	
Survivors	14–26	1 × 10^6^–1 × 10^9^	High load
	26–29	1 × 10^3^–1 × 10^6^	Medium load
	29–33	10–1 × 10^3^	Low load
	33–36	<10	Non-carrier

* *Vp* qPCR using DNA from HP of dead and survivor animals challenged with *Vp*_AHPND_. HP were weighted before the DNA extraction and the bacterial load was normalized according to the HP size.

**Table 6 animals-13-01354-t006:** Bacterial load in hepatopancreas (HP) of survivor shrimps after challenges with *Vibrio parahaemolyticus* (*Vp*_AHPND_ strain).

Challenge with *Vp*_AHPND_ after	Diet	CT Range (*Vp* qPCR)	*Vibrio parahaemolyticus* (CFU/g HP)	Carrier Category
4-week feeding schedule	G (control)	26.5–36.59	10–8 × 10^5^	Low-medium
	E (functional)	15.42–27.5	8 × 10^5^–8 × 10^8^	Medium-high
	F (functional)	25.5–32.8	10–1 × 10^7^	Low-medium
5-week feeding schedule	G (control)	16.5–30.5	1 × 10^2^–9 × 10^9^	Low-high
	E (functional)	13.5–30.18	10–1 × 10^9^	Low-high
	F (functional)	26.2–33.1	10–1 × 10^6^	Low-medium

**Table 7 animals-13-01354-t007:** Detection of *Vibrio parahaemolyticus* by *Vp* qPCR assay in water of tanks with challenged shrimps. CT range and equivalencies in CFU/mL in water samples collected from maintenance tanks at different post-challenge times.

Days Post-Challenge	CT Range (*Vp* qPCR)	*Vibrio parahaemolyticus* (CFU/mL Water)
0 (before infection)	34–35	0
1	22–25	5 × 10^7^–3 × 10^8^
2	21–29	5 × 10^3^–5 × 10^8^
3	23–29	5 × 10^3^–5 × 10^8^

## Data Availability

The data and models that support the study findings are available from the corresponding authors upon reasonable request.

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
