# Peer review of "Exploring the Effect of Functional Diets Containing Phytobiotic Compounds in Whiteleg Shrimp Health: Resistance to Acute Hepatopancreatic Necrotic Disease Caused by Vibrio parahaemolyticus"

_animals, 2023, doi:10.3390/ani13081354_

Round 1

Reviewer 1 Report

The study evaluated the impact of functional diets containing phytobiotic compounds on the resistance of whiteleg shrimp to Vibrio parahaemolyticus by detecting the antibacterial activity of additives in vitro, as well as the mortality, percentage of carriers, and bacterial load in the hepatopancreas of whiteleg shrimp after bacterial challenge tests. The experimental design is rigorous and reasonable, and the research results have important significance for promoting the healthy cultivation of whiteleg shrimp. However, there are still some points in the article that need to be improved. Comments and suggestions are as follows:

1. In the article, “PirAB” should be modified to “PirA and PirB”; in line 74, “this” should be modified to “these”; in line 83, “only VpAHPND strains were the responsible for” should be deleted “the”; in line 163, “0.8 g of Additive F/Kg” should be modified to “0.8 g of additive F/Kg”; “V. parahaemolyticus” in line 309 and line 314, and “Vp” in line 481 should be italicized. There are many other minor problems in the article, please pay attention to checking and modifying.

2. Some sentences in the text need to be improved to make their expression clearer and more accurate. For instance, in line 116-117, “assess their effects on the resistance of the species to the AHPND”; in line 119-120, “we estimated …… the bacterial loads in survivors by using an optimized qPCR from HP”.

3. In line 174-75, “one group fed CTRL diet and the other groups fed functional diets (E and F) for 4-5 weeks with a daily regimen of around 10%”, 10% of what? Is of the shrimp's weight?

4. In line 208, “Shrimps were fed the same diets along the post-challenge period”, which diets? Control diet or others?

5. In Table 4, why not use the same amount of experimental individuals?

6. Figure 3 shows the survival rate. According to the description in the text, it is best to use mortality data for plotting.

7. Why design two times for bacterial challenge tests?

8. In Discussion, this section should mainly discuss the study results. However, the content in line 421-462 is similar and repetitive to the content in Introduction section. It is recommended to integrate this content into the Introduction section.

9. In line 487-488, “but we assessed another approach, the V. parahaemolyticus loads in HP of survivors after induced infection”. Please explain why bacterial load in HP of survivor shrimps reached the highest value in diet E group, while percentage of carrier shrimps in diet E group was lowest.

10. In line 496-502, is it possible that the V. parahemolyticus in water originate from the surface of the shrimp after bathing challenge rather than being released by the infected shrimp?

11. In line 519-520, “Moreover, this feeding strategy also promoted the reduction of the percentage of carriers after the induced infection as well as the bacterial load in hepatopancreas of survivors”. Results showed that the bacterial load in hepatopancreas of survivors in diet E group did not decrease.

12. In lines 506-508, It is recommended to deleteOverall, our results show evidence that diet supplemented with essential oils from thyme and cinnamon, administered for 4 weeks, would be a promising practice to improve health of shrimps and mitigate the negative effects of AHPND”, because it is mentioned in the Conclusions section.

Author Response

Dear Reviewer 1, thank you very much for your deep review to our manuscript and for your valuable comments.

Point 1: In the article, “PirAB” should be modified to “PirA and PirB”; in line 74, “this” should be modified to “these”; in line 83, “only VpAHPND strains were the responsible for” should be deleted “the”; in line 163, “0.8 g of Additive F/Kg” should be modified to “0.8 g of additive F/Kg”; “V. parahaemolyticus” in line 309 and line 314, and “Vp” in line 481 should be italicized. There are many other minor problems in the article, please pay attention to checking and modifying.

Response 1: We have improved the article following your suggestions. Other minor problems have been also modified and can be followed on the “tracked-changes” version of the manuscript, attached to this message.

Point 2: Some sentences in the text need to be improved to make their expression clearer and more accurate. For instance, in line 116-117, “assess their effects on the resistance of the species to the AHPND”; in line 119-120, “we estimated …… the bacterial loads in survivors by using an optimized qPCR from HP”.

Response 2: We agree with your suggestions. The senteces have been changed as follows: line 119-120, “administered to shrimp to evaluate their putative benefitial effect on the resistance of the animals to the AHPND pathology”; line 122-123, “we calculated the percentage of carriers and the bacterial load in HP of survivors by using an optimized qPCR protocol”.

Point 3: In line 174-75, “one group fed CTRL diet and the other groups fed functional diets (E and F) for 4-5 weeks with a daily regimen of around 10%”, 10% of what? Is of the shrimp's weight?

Response 3: Yes, it is 10% of the shrimps’ weight. We have clarified it on the text (line 187-188).

Point 4: In line 208, “Shrimps were fed the same diets along the post-challenge period”, which diets? Control diet or others?

Response 4: Each group was fed the same diets (either CTRL or functional) during the post-challenge, as during the feeding trial. We agree that sentence could be confused, thus we have included this clarification on the text (line 222-223).

Point 5: In Table 4, why not use the same amount of experimental individuals?

Response 5: Dear reviewer, unfourtunatly we had problems with the animals management (cannibalism events) in some groups prior to the bacterial challenge. Therefore we had to readjust the number of experimental individuals.

Point 6: Figure 3 shows the survival rate. According to the description in the text, it is best to use mortality data for plotting.

Response 6: Thank you for pointing out this. We aimed to present the results in two different graphical formats, Figure 3 represents the evolution of mortality and final percentage of survival while Figure 4 represents the final mortality. We understand it might be confused according to the description in the text, therefore we have modified the Figure 3 title and legend for clarification.

Point 7: Why design two times for bacterial challenge tests?

Response 7: We included two times for bacterial challenges tests in order to test if longer feedings with functional diets would improve the health status of the animals. One week feeding more with functional diets is an affordable economical invest for the industry and we wanted to evaluate if there was a benefit in longer feedings.

Point 8: In Discussion, this section should mainly discuss the study results. However, the content in line 421-462 is similar and repetitive to the content in Introduction section. It is recommended to integrate this content into the Introduction section.

Response 8: Following your suggestion we have shortened and integrated some parts of the first part of the discussion into the introduction. We hope you appreciate our changes (see tracked-changes document).

Point 9: In line 487-488, “but we assessed another approach, the V. parahaemolyticus loads in HP of survivors after induced infection”. Please explain why bacterial load in HP of survivor shrimps reached the highest value in diet E group, while percentage of carrier shrimps in diet E group was lowest.

Response 9: We appretiate you have noticed this aspect. We consider that the most important parameter in our study is the decrease of the percentage of carriers in groups fed functional diets which is consistent with the higher survival observed in animals fed diet E. It is true that the baterial load values are higher in survivors of this group, and are similar to those found in dead animals in groups fed other diets. Thus, we believe that our results suggest that animals fed this diet can survive with higher bacterial loads, which is also a clear benefit. We do not know the causes and we agree it would be interesting to assess this in future studies (i.e. with longer follow-up on survivors to see if they end up eliminating V. parahaemolitycus cells from HP).

Point 10: In line 496-502, is it possible that the V. parahemolyticus in water originate from the surface of the shrimp after bathing challenge rather than being released by the infected shrimp?

Response 10: We believe that this is possible during the first 24 h post-challenge, when the V. parahaemolyticus loads in water are higher, and they could partially come from the surface of the challenged shrimps. After that, the baterial loads decrease and we doubt the bacteria remain on the shrimp’s surface. Nevertheless, we have not done any studies on the bacterial loads on shrimp’s surface and we cannot provide further data on this.

Point 11: In line 519-520, “Moreover, this feeding strategy also promoted the reduction of the percentage of carriers after the induced infection as well as the bacterial load in hepatopancreas of survivors”. Results showed that the bacterial load in hepatopancreas of survivors in diet E group did not decrease.

Response 11: Thanks for the comment. We agree with you and the sentence has been modified in line 614-615, as following: “Moreover, this feeding strategy also promoted the reduction of the percentage of carriers after the induced infection”.

Point 12: In lines 506-508, It is recommended to delete“Overall, our results show evidence that diet supplemented with essential oils from thyme and cinnamon, administered for 4 weeks, would be a promising practice to improve health of shrimps and mitigate the negative effects of AHPND”, because it is mentioned in the Conclusions section.

Response 12: We have followed your recommendation and modified the paragraph that remains in line 604-606, “It is important to remark that we have adapted the time of functional diet administration to the highly risky period for disease outbreaks in farms, often within 30-35 days after introducing post-larval or juvenile shrimp into contaminated tanks. Nevertheless, further studies are needed to assess the effectiveness of the additive dose and the feeding regime used here when scaling to shrimp farm conditions.”

Reviewer 2 Report

The article is well written and has a lot of relevant information, an extensive and important bibliographic survey on the subject addressed.

Despite this, the results are not very expressive in showing the real effectiveness of essential oils in increasing shrimp production. Or maybe they are not so evident when analyzing the results in more detail.

One factor to consider is that animals weighing 1 g had poor survival. When thinking about shrimp production, this is the maximum size for growing storage, as it would be the ideal size for animals that go through a long nursery. Animals weighing 2 g are usually already in grown in farms. 

However, the protocol used and the analyzes carried out provide important information in determining parameters and methodologies for conducting studies of this nature and can easily be replicated in other studies in this line of research. Perhaps this is the most relevant part of the paper.

There is a lack of information on experimental management and control of water quality parameters and their maintenance. Due to the mortalities presented without any logical explanation, it would be good to have these data to know if there could be some kind of relationship between the total loss of animals in a tank and the quality of the water in the experiment.

The first part of the discussion makes no reference to the work and is rather a repetition of the introduction. I would consider shortening or going directly to the observed results, as I could go deeper into the specific topic of the article.

Author Response

Dear Reviewer 2, thanks for your kind review and comments.

Point 1: One factor to consider is that animals weighing 1 g had poor survival. When thinking about shrimp production, this is the maximum size for growing storage, as it would be the ideal size for animals that go through a long nursery. Animals weighing 2 g are usually already in grown in farms.

Response 1: Regarding the animal weight, 1 g shrimps were used in a pre-challenge with few individuals to optimise the infection model and adjust the lethal dose 50% (LD50). Those animals did not feed the functional diets. Although it seems to be a tendency of more poor survival in 1 g animals, this cannot be generalised due to the low number of experimental individuals used in the pre-challenge. As we stated in the last paragraph of Discussion, large scale trials should be done to assess this aspect.

Point 2: However, the protocol used and the analyzes carried out provide important information in determining parameters and methodologies for conducting studies of this nature and can easily be replicated in other studies in this line of research. Perhaps this is the most relevant part of the paper.

There is a lack of information on experimental management and control of water quality parameters and their maintenance. Due to the mortalities presented without any logical explanation, it would be good to have these data to know if there could be some kind of relationship between the total loss of animals in a tank and the quality of the water in the experiment.

Response 2: Water quality parameters (pH, temperature, salinity, nitrogen compounds) were controlled daily and no deviations were observed in any of the experimental tanks. We have added these lines in the text for clarification. We did not carry out any further tests and cannot confirm the presence of any toxic debris that could be involved in the observed high mortalities.

Point 3: The first part of the discussion makes no reference to the work and is rather a repetition of the introduction. I would consider shortening or going directly to the observed results, as I could go deeper into the specific topic of the article.

Response 3: Following your suggestion we have shortened and integrated some parts of the first part of the discussion section into the introduction one. We hope you appreciate our changes (see tracked-changes document attached to this message).
